# Effects of Ultrasound-Assisted Vacuum Impregnation Antifreeze Protein on the Water-Holding Capacity and Texture Properties of the Yesso Scallop Adductor Muscle during Freeze–Thaw Cycles

**DOI:** 10.3390/foods11030320

**Published:** 2022-01-24

**Authors:** Yuyao Shi, Hongli Wang, Yao Zheng, Zehui Qiu, Xichang Wang

**Affiliations:** 1College of Food Science and Technology, Shanghai Ocean University, Shanghai 201306, China; m190310917@st.shou.edu.cn (Y.S.); d190300066@st.shou.edu.cn (H.W.); d180300058@st.shou.edu.cn (Y.Z.); m200300793@st.shou.edu.cn (Z.Q.); 2Shanghai Engineering Research Center of Aquatic-Product Processing & Preservation, Shanghai 201306, China

**Keywords:** scallop adductor muscle, antifreeze protein, vacuum impregnation, ultrasound, water-holding capacity, texture properties

## Abstract

The effect of antifreeze protein (AFP) on the water-holding capacity (WHC) and texture properties of the *Patinopecten yessoensis* adductor muscles during freeze–thaw cycles (FTCs) were evaluated based on three impregnation methods: general impregnation (GI), vacuum impregnation (VI), and ultrasound-assisted VI (US-VI). The WHC, texture properties, and tissue microstructure were all evaluated. Results showed that the WHC and texture properties of adductor muscle were significantly improved in the VI and US-VI groups during FTCs (*p* < 0.05). The WHC of the adductor muscle in the US-VI group was maximally enhanced in terms of yield (6.63%), centrifugal loss, cooking loss, and T_22_. The US-VI group of the adductor muscle had the optimal chewiness and springiness compared to others, and the shear force and hardness were most effectively enhanced by VI. The growth and recrystallization of ice crystals in the frozen adductor muscle were significantly inhibited by VI and US-VI. The average cross-sectional area and roundness of ice crystals in the US-VI group were decreased by 61.89% and increased by 22.22% compared with those of the control, respectively. The partial least squares regression (PLSR) model further confirmed that the WHC and texture properties of the adductor muscle were correlated appreciably with the degree of modification of ice crystal morphology through the AFP.

## 1. Introduction

Yesso scallop (*Patinopecten yessoensis*), a mariculture bivalve, is one of the most important aquaculture species in Asian nations [1]. Scallops were widely distributed along the coastline of Northern Japan, the far east of Russia, the Korean Peninsula, and Bohai Bay of China, which were highly prized for their rapid growth and high nutritional value [2]. It has been reported that, from 2012 to 2020, the production of scallops showed an increasing trend, with total scallop production reaching 1.75 million tons in 2020 (China fishery statistical yearbook). Live scallops show great profitability but are highly susceptible to deterioration and having a short shelf-life. Freezing could prolong the shelf-life of products and maximize the preservation of their original properties by inhibiting microbial activity and enzymatic reactions [3]. However, long-term frozen storage inevitably leads to ice crystal growth [4], which causes irreversible mechanical damage to the muscle fibers, thus leading to a decrease in WHC and texture properties [5,6], and indirectly triggering protein denaturation [7] and lipid oxidation [8], ultimately affecting the sensory experience of consumers and commodity value [9]. Therefore, the WHC and texture properties dominate the degree of deterioration of muscle quality.

Antifreeze agents could decrease the freezing point, inhibit the growth of ice crystals, and thus prolong the shelf-life of frozen products. The addition of cryoprotectants to aquatic products has been found to be one of the most effective ways of alleviating the deterioration of quality during freezing and frozen storage [10,11]. Compared with other common antifreeze agents, antifreeze proteins (AFPs) had better frozen stability [12]. AFPs refer to specific peptides or glycopeptides with heat hysteresis (TH) activity that are used by organisms to adapt to extreme cold conditions and protect them against freezing damage [13]. AFPs, as the most typical ice-active ice blocker, are adsorbed to the surface of ice crystals through hydrogen bonds and accumulate at the ice–water interface, inhibiting the formation, growth, and recrystallisation of embryonic ice crystal through the Kelvin effect. Interestingly, AFPs are still effective in reducing ice crystal damage to cells and tissues in the freezing process, even at low concentrations (0.1 g/100 mL or lower) [14]. However, appropriate external forces are required to further disperse and permeate into the tissue due to the large molecular weight of AFP [12]. Research on AFPs has mainly been aimed to the preparation and evaluation of AFP [10], the effect of protein oxidation [12], and the protein aggregation state [15], but little attention has been paid to the WHC and texture properties of muscle.

Vacuum impregnation (VI), a promising nonthermal technique, could facilitate the bidirectional mass transfer between the capillary structure of foods and the impregnating solution due to the pressure difference [16]. It has been found to effectively introduce salt [17], calcium and black carrot phenolics [16], fish gelatin, and grape seed extract [18] into tissues. However, studies conducted on the addition of AFP to tissues through the VI are rare. Ultrasound treatment refers to an innovative technology that does not destroy the tissue structure, and its sound range is divided into high-frequency low-intensity (>100 kHz, <1 W/cm^2^) and low-frequency high-intensity (20~500 kHz, >1 W/cm^2^) [19]. High-intensity ultrasound treatment, used as an initial pretreatment method for food preservation, can increase the permeability of muscle tissues through cavitation forces and accelerate the rate of mass transfer, which facilitate the ripening, drying, and tenderizing stages of meat processing [20,21]. Furthermore, recent studies have confirmed that ultrasound treatment can improve the diffusion of salt [17] and the WHC [22].

Although VI and ultrasound treatment have been confirmed to improve the permeability, there has been scarce information on the effects of ultrasound combined with vacuum impregnation AFP, especially the comprehensive investigation of quality on the scallop adductor muscle after the combination. Thus, based on the US-VI, the effect of AFP on the WHC and texture properties of the scallop adductor muscle were investigated. Some reference for the quality control of frozen aquatic products under the effect of the AFP during storage could be provided.

## 2. Materials and Methods

### 2.1. Sample Preparation

The Yesso scallops (*Patinopecten yessoensis*) (weight: 148.61 ± 19.68 g, *n* = 300) were provided by Dalian Zhangzidao Group. The fresh scallops were transported back to the laboratory at 4 °C conditions. The adductor muscles had an average net weight of 16.47 ± 3.44 g after the gills, viscera, gonads, and cortical membrane were removed (Figure 1A). The AFP (purity greater than 95%) was purchased from Anfei Bio. (Nanjing, China). The THA of this AFP was nearly 1.5 °C. Other biochemical reagents were at least of analytical grade.

The fresh adductor muscles of 300 scallops were randomly divided into four groups. The respective group was pretreated using the following methods: (1) Control: placed in a refrigerator at 4 °C for 50 min; (2) GI: general impregnated in 0.2 g/L AFP solution; (3) VI: vacuum impregnated (0.07 MPa) in 0.2 g/L AFP solution; (4) US-VI: ultrasound-assisted (43 kHz, 200 W/cm^2^) vacuum impregnated in 0.2 g/L AFP solution. The adductor muscles of the AFP groups above were all impregnated at 4 °C for 40 min at a material–solution ratio of 1:1.5, and subsequently impregnated under atmospheric pressure for 10 min. Figure 1B illustrates the experimental design and potential mechanism. The procedure, freezing the adductor muscles for 1.5 h in the air blast freezer (BU15, Proton-chef, China), storing at −20 °C for one weak, followed by thawing at 4 °C for 3.5 h until their core temperature reached 0~4 °C as measured by Thermo Recorder RX6000C (T&D, Ltd., Beijing, China), was repeated 1, 3, and 5 times. A total of 30 samples were randomly selected from each subgroup for analysis, and the remaining samples were kept at −20 °C for a further week until the next FTC.

### 2.2. Methods

#### 2.2.1. Yield and Moisture

The yield was measured according to Jiang, Nakazawa, Hu, Osako, and Okazaki [7], which was determined by the mass ratio of the samples after thawing (M_2_) to those before impregnating (M_1_) (also considered as the fresh sample) by Formula (1):(1)$${\mathrm{Yield}\;(\%)=M}_{2}{/M}_{1}\text{×}100\%$$

Each group was repeated at least 12 times.

The moisture was performed using the direct drying method in accordance with the National Food Safety Standard GB 5009.3-2016 in China. 

#### 2.2.2. Centrifugal Loss and Cooking Loss

The samples (2.00 ± 0.02 g) were cut down along the direction of the thawed scallop adductor muscle fiber and marked as M_3_. After being centrifugated at 3821× *g* (H1850R, Changsha, China) for 10 min at 4 °C, the samples were weighed again (M_4_). The result was obtained from Formula (2):(2)$$\mathrm{Centrifuging}\;\mathrm{loss}\;\left(\%\right)=\frac{M_{3}–M_{4}}{M_{3}}\text{×}100\%$$

The cooking loss was measured according to Vaskoska, et al. [23] with some modifications. The thawed adductor muscle (M_4_) was placed at the bottom of a polyethylene bag with marbles to prevent the sample from floating. Next, the sample was placed in a 90 °C water bath (Model 9112T12E, PolyScience, Niles, IL, USA) and then heated for 15 min. The weight of the cooked sample was recorded as M_5_. The formula was as follows:(3)$$\mathrm{Cooking}\;\mathrm{loss}\;\left(\%\right)=\frac{M_{4}{\text{–}M}_{5}}{M_{4}}\text{×}100\%$$

#### 2.2.3. Texture Profile Analysis (TPA) and Shear Force

The texture profile analysis (TPA) and the shear force were measured through the TA.XT Plus texture analyzer (Stable Micro Systems Ltd., Godalming, UK) as described by Li, et al. [24] with some modifications. The samples were sliced into 2 × 2 × 1.5 cm^3^ approximate cubes. The parameters of TPA mode with a 50-mm diameter cylindrical probe were as follows: the pretest speed was 1.0 mm/s, the test speed was 1.0 mm/s, the trigger force was 5 g, and the test interval was 5 s, followed by a two-cycle compression test to 50% of its original height. The parameters of shear force with a knife blade (HDP-BSW) were as follows: the pretest speed was 5.0 mm/s, the test speed was 1.0 mm/s, the post-test speed was 5.0 mm/s, and the distance was 40 mm, as well as the trigger force of 5 g. The sample was cut in the direction perpendicular to the muscle fibers. At least 6 replicates were detected for each group.

#### 2.2.4. Low-Field Nuclear Magnetic Resonance (LF-NMR) and Magnetic Resonance Image (MRI) Measurement

T_2_ transverse relaxation measurements and MRI were performed on an LF-NMR Analyzer PQ001 (MesoMR23-060H.I, Niumag Corporation, Shanghai, China) with 0.5 T permanent magnet corresponding to a proton resonance frequency of 20 MHz. Samples were placed in NMR tubes and the transverse relaxation time (T_2_) was determined with the CPMG pulse sequence. Acquisition parameters were as follows: SW = 100 kHz, PRG = 1, TW =3000 ms, P2 = 37.00 μs, TE= 0.30 ms, NECH = 6000, and NS = 8. 

The proton density pseudo-color images of the samples were obtained by MRI imaging software, as well as MSE spin-echo imaging sequence acquisition as described by [25] with some modifications. The following scanning protocols were used: field of view of 80 × 80 mm, slice width of 3.0 mm, slice gap of 2.5 mm, read size 256, phase size 192, T_1_ weighted image echo time (TE) of 18.124 ms, repetition time (TR) of 160 ms, T_2_ weighted image echo time (TE) of 50 ms, and repetition time (TR) of 1600 ms. 

#### 2.2.5. Observation and Determination of Ice Crystal Morphology

Changes in ice crystal morphology of the scallop adductor muscle were determined using the hematoxylin eosin staining method described by [7] with some modifications. The samples were fixed, dehydrated, and embedded in paraffin. The microstructural variations of the transverse and longitudinal sections (5 μm in thickness) were observed separately under a light microscope (Eclipse E200 Biological Microscope, Nikon Instruments Ltd., Tokyo, Japan). Image pro plus 6.1 software (Media Cybernetics, Silver Spring, MD, USA) was adopted to evaluate the average equivalent diameter, cross-section area, roundness, and elongation of ice crystals based on the transverse optical micrographs. To confirm the accuracy of the data, at least three photos from each group were chosen for analysis with more than 300 ice crystals.

The elongation was considered the ratio of the major axis length to the minor axis length. If the value of elongation was 1, the ice crystal studied would be roughly square or circular. As the value increased from 1, the ice crystal becomes elongated. The roundness (R) was calculated according to Luan, et al. [26] as follows:(4)$${R=P}^{2}/4\pi A$$
where A is the cross-sectional area and P represents the perimeter. The value of roundness 1 represents a circle. A smaller value represents a rounder object. For each case considered, over 100 ice crystals were analyzed. The areas (defined as A) of the ice crystals (white areas in the images) were determined by outlining the ice crystals using the Count and Size feature.

#### 2.2.6. Scanning Electron Microscopy (SEM) Observation

The microstructural changes in the muscle fibers were observed using the method described by [11] with some modifications. The thawed adductor muscle was cut into 2–3-mm-thick pieces, fixed in 2.5% glutaraldehyde solution at 4 °C for 12 h, then rinsed 3 times with 0.2 mol/L phosphate buffer, pH 7.2, for 15 min each time. The samples were dehydrated in a gradient of 70%, 90%, 95% (*v/v*) ethanol solution and anhydrous ethanol, all for 30 min. Finally, the samples were rinsed with ethanol:tert-butanol solutions of 3:1, 1:1, and 1:3 by volume for 10 min each time. After freeze-drying, the samples were coated and placed under an SU5000 scanning electron microscope (HITACHI, Tokyo, Japan) for observation.

#### 2.2.7. Statistical Analysis

All experiments in this study were performed at least in triplicate. Statistical analysis of the results was carried out by using the SPSS 21.0 (SPSS Inc., Chicago, IL, USA) software, followed by Duncan’s test, which conformed to the normal distribution and was expressed as mean values ± standard deviations (SD). Comparison of treatment averages was performed by ANOVA test, and a *p*-value of less than 0.05 indicated that the means differed significantly. All graphs were drawn using Origin 2018 (Origin Lab Corp, Hampton, VA, USA). The PLSR model was performed using SIMCA software (MKS UMETRICS, version 14.1).

## 3. Results and Discussion 

### 3.1. The Effects of AFP on the Water-Holding Capacity

#### 3.1.1. Changes in WHC

Water-holding capacity (WHC) refers to how structural changes in meat affect its water distribution and mobility, as well as the ability to maintain its own water and added water during ageing [27], which is a key criterion for the assessment of meat quality [28]. After FTC-1, the yield was greater than 100%, except for that of the control, and the magnitude of yield complied with the order of US-VI > VI > GI > Control (*p* < 0.05) (see Table 1). The cooking loss and centrifugal loss in US-VI group was increased by 45.76% and 34.52% compared to the control group, separately. The reduction in cooking loss enhanced the juiciness of the adductor muscle while increasing the profitability of the restaurant. The impregnated weight gain of the adductor muscle was greater than the thawing loss in the AFP groups, and more impregnating solution was allowed to hold in the adductor muscle by the VI and ultrasound treatment. It was speculated that the gas, free water, and soluble substances were partially discharged by mutual mass transfer between the food capillary structure and the impregnated solution due to the pressure differences, which made the impregnating solution tend to penetrate into the tissue [29]. When subjected to multiple FTCs, the yield of all samples decreased markedly, which was also observed in unsalted tuna meat, as reviewed by Jiang, Nakazawa, Hu, Osako, and Okazaki [7]. After five FTCs, the highest yield was in the US-VI group, 6.63% higher than that of the control group, further suggesting that the WHC of the adductor muscle was maximally enhanced by the US-VI treatment. The cavitation force made the solvent penetrate deeper into tissues, thereby facilitating mass transfer, which is thought to be responsible for the increase in WHC caused by ultrasound [16,30]. The variation in moisture showed a similar trend to the yield among the groups induced by FTCs, whereas no significant difference was found in numerical values.

#### 3.1.2. Changes in Water States and Distribution

Low-field nuclear magnetic resonance (LF-NMR) and magnetic resonance imaging (MRI) have been found to be an effective, noninvasive alternative to characterize the water dynamic of muscle during processing and storage in recent years [25,31]. LF-NMR transverse relaxation and the MRI analysis were conducted to test the state of water distribution and transformation of the scallop adductor muscle during repeated FTCs. Figure 2A presents the distribution of T_2_ relaxation times calculated by fitting the CPMG pulse sequence. T_2b_ (0–10 ms), the proton pool with the shortest relaxation time, represents bound water attached on the polar groups of macromolecules directly. T_21_ (10–100 ms) and T_22_ (100–1000 ms) were assigned to immobilized water entrapped within the myofibrillar network and ascribed to free water in the myofibril lattice, respectively. During the FTCs, T_2b_ and T_21_ did not change significantly in any treatment group (*p* > 0.05), indicating that the bound water was associated with macromolecule and could not easily migrate to other components. 

However, T_22_ increased significantly with an increasing number of FTCs (*p* < 0.05); a similar behavior of T_22_ was reported in instant sea cucumber subjected to multiple FTCs [25]. Intra-myofibrillar water gradually migrated to the extracellular space and formed larger and irregular ice crystals as a result of intracellular osmotic flow and water redistribution (Lebovka, Bazhal, & Vorobiev, 2001), thereby causing considerable mechanical damage to cell membranes and myofibrillar proteins, and thus reducing the amount of water that could actually be reabsorbed by the myofibrils upon thawing (Sánchez-Alonso et al., 2014; Xia, Kong, Liu, & Liu, 2009). In contrast, the VI and US-VI groups significantly inhibited the changes in T_22_, weakened the binding of water and muscle fibers, and improved the WHC of the adductor muscle.

Due to the proton density T_2_ weighted images derived from MRI, the water distribution in the scallop adductor muscle could be visualized. The corresponding pseudo-color images are shown in Figure 2B, in which red represents high proton signal density, and the signal density of yellow and blue decreased in that order. The gradual increase in the image brightness from the outside to the inside of each picture indicated that water was drained from the edges of the scallop adductor muscle as the drip loss through channels formed by the inter-myofibrillar spaces and, hence, there was a higher moisture content in the central region. As the number of FTCs increased, the red areas in the scallop adductor muscle were gradually replaced by yellow, and the image brightness of the US-VI and VI groups was higher than that of the others. This result might be due to the fact that AFP inhibited the growth and recrystallisation of ice crystals, which reduced the extent of disruption of muscle fibers, and thus led to a better WHC. The variation of the moisture reflected by the brightness of the spectrum showed the same trend as the variation in the yield for each group in Table 1, further confirming that the US-VI might have the optimal WHC.

### 3.2. The Effects of AFP on the Texture Properties

Texture properties are crucial indicators to assess the quality of frozen aquatic products, which are mainly related to the degree of disruption of muscle fibers and subsequent protein structure [32]. Texture degeneration will occur when large and irregularly shaped ice crystals are formed during freezing, thereby causing mechanical damage to cell membranes and connective tissues [33]. As shown in Figure 3, the hardness, chewiness, springiness, and shear force all showed a declining trend with an increasing number of FTCs. The texture properties of the VI and US-VI groups were much closer to those of fresh samples (control group of FTC-0) and outperformed those of the control and GI group, irrespective of the number of FTCs (*p* < 0.05). The chewiness and springiness in the US-VI group of the adductor muscle were higher than those of the VI group. Moreover, it is also noteworthy that the shear force and hardness of the US-VI group in FTC-1 were markedly lower than those of VI group (*p* < 0.05), probably because ultrasound had a tenderizing effect on muscles while increasing the permeability of muscle tissues. After five FTCs, the shear force of the VI and US-VI groups was 49.46% and 44.18% higher than that of the control group, respectively. The observed decline in the shear force showed a similar result as the 70.0% decline in mirror carps when subjected to five FTCs [34]. It has been hypothesized that the growth and recrystallization of ice crystals were inhibited by the AFP, a phenomenon that reduces the damage to myofibrils and connective tissue membranes by ice crystals and slows down the degradation of muscle fiber structures, resulting in increased tissue firmness. The above results fully confirmed that the chewiness and springiness of adductor muscle could be greatly improved by US-VI, and that shear force and hardness were more effectively enhanced through the VI.

### 3.3. The Effects of AFP on the Microstructure

#### 3.3.1. Changes in Ice Crystal Morphology

The results of the histological examination (H&E staining) of the ice crystal morphology of the adductor muscle observed by the light microscope are presented in Figure 4a of a transverse section and Figure 4b of a longitudinal section, respectively. Ice crystals formed during the freezing process affected cell integrity, and the evolution of ice crystals determined the extent of tissue recovery and quality deterioration after thawing. In general, large and irregularly shaped ice crystals caused irreversible damage to the cell structure, thereby resulting in a reduction in WHC, texture properties, and nutrients, whereas smaller size and spherical ice crystals reduced the mechanical damage to muscle fibers [35,36]. After FTCs, the ice crystals in the VI and US-VI groups were smaller, round or elliptical in shape, with neatly arranged fiber bundles, thereby causing less damage to cell structure compared with the control group. These results are generally in accordance with the findings in ice crystal morphology of mirror carp caused by FTCs [36]. The channels for drip loss in the control group were formed by the expanded extracellular space and disintegration of connective tissue, which may be the major factor of the reduction in WHC. The hydrophilic surface of the AFP side chain is thought to match and bind to the prismatic surface (growth surface) of the ice crystal through hydrogen bonding, exposing the hydrophobic surface with a high surface free energy, and thereby lowering the freezing point and hindering the growth of ice nucleation and recrystallization [10].

The ice crystal morphology of the muscle tissue in cross-sectional microscopic images was further analyzed using Image analysis software (Image-Pro Plus 6.0, Media Cybernetics, American). Figure 4c indicates that the average cross-sectional area, equivalent diameter, and roundness of the ice crystals formed in the tissues were significantly affected by the repeated freezing–thawing, but no significant difference in the elongation. Compared with the GI, the VI and US-VI groups showed a significantly inhibited growth of ice crystals (*p* < 0.05) and greater roundness of ice crystals, which demonstrated that the ice crystal morphology was effectively modified through the VI and US-VI treatments. What is more noteworthy is that the average cross-sectional area of ice crystals in the US-VI group (1254.13 ± 107.32 μm^2^) was decreased by 61.89% compared to the control group (3291.10 ± 251.80 μm^2^) after five FTCs. The same trend was observed in roundness. The increase in the ice crystal area during the FTCs was probably ascribed to the incorporation of crystals from adjacent areas to form a larger crystal, which was a melt–diffuse–recrystallization mechanism known as Ostwald ripening [37]. AFP at the ice–solution interface produces the ice surface curvature effect on the ice surface due to the adsorption-inhibition mechanism, thus limiting the growth and incorporation of ice crystals in the adductor muscle and modifying the ice crystal morphology [38,39], which further verified that US-VI has a positive modification effect on the ice crystal morphology. The present results are consistent with results of the effect of xylooligosaccharide (XO) on ice crystal morphology of frozen peeled shrimp exposed to temperature fluctuations [11]. It confirmed that AFP can be used as an alternative to carbohydrate-based cryoprotectant in aquatic products, reducing calorie intake and making it more accessible to a broader variety of consumers [40].

#### 3.3.2. Changes in SEM Observation

Up to 85% of the water in muscle cells is occupied by myofibrils, which is maintained by capillary forces arising from the arrangement of thick and thin filaments in the myofibrils. Therefore, the key factors affecting drip loss are the contraction of muscle fibers and the expansion of extracellular space [41]. Figure 5 shows the transverse SEM image of the tissue microstructure of the adductor muscle. The interfibrillar holes were thought to be the irreversible deformation resulting from the deformation or even breakage of myofibers within the tissue due to ice crystal growth [42]. The fiber bundles of the fresh sample were well arranged with small and even distribution of pores, showing a tight honeycomb structure (see Figure 5). The intracellular holes continued to become larger as the number of FTCs increased, resulting in a significant separation of connective tissue and muscle bundles in all groups. Larger holes and irregularly shaped ice crystals were visible in the control group, which caused severe distortion and deformation of muscle fibers. The pores in the VI and US-VI groups were smaller, and circular or elliptical in shape, rather than the ice columns observed in the control group. These results are generally in accordance with the findings in largemouth bass during FTCs by herring AFP [12]. It was then speculated that the combination and accumulation of AFP and ice crystals at the ice–water interface led to an increase in the curvature of the ice crystals and a consequent expansion in surface area and vapor pressure at the edges, and thereby changing the equilibrium of the system and lowering the freezing point [43]. From the aspect of tissue microstructure, the squeezing effect of ice crystals caused the breakage of myofibrils and connective tissues, weakening of mechanical strength, which, in turn, led to the diminished WHC and muscle fibers, and the loss of the released cellular substances as the thawing drip was dropped. Thus, the WHC and texture properties of the adductor muscle were considerably improved by the VI and US-VI.

### 3.4. The Partial Least Squares Regression (PLSR) Model

The relationship between WHC, texture properties, and ice crystal morphology, which was based on the evaluation of the degree of quality deterioration of the adductor muscle with different impregnation methods under freeze–thaw cycles, was further explored. Figure 6 illustrates a partial least squares regression (PLSR) model with six WHC-related indicators, four texture properties indexes as the X-variables and ice crystal morphology indicators as the Y-variables, which was used to analyze the correlation between the first two components (t1, t2). The PLSR was used to define the relationship between the WHC, texture properties, and ice crystal morphology because the effect of AFP is characterized by the degree of variation in ice crystal morphology, mainly including the average cross-section area, equivalent diameter, and roundness of ice crystals. The variables around the edge of ellipse were well interpretated by the model. Similar results were found for the average equivalent diameter, cross-section area, T_22_, and cooking loss, all of which were negatively correlated with the roundness of ice crystals, hardness, springiness, chewiness, and shear force along the factor-1 axis, respectively. A smaller equivalent diameter of ice crystals could lead to a greater parameter of texture and shorter T_22_, confirming that the degree of modification of ice crystal morphology by the AFP appreciably affects the WHC and textural properties of the muscle.

## 4. Conclusions

Overall, this study confirmed that, in addition to a certain degree of tenderness, the WHC and ice crystal morphology of the adductor muscles could be maximally improved by the combination of ultrasound treatment and VI during FTCs. As indicated by the result of the PLSR model, the ice crystal morphology effected by the AFP was strongly correlated with the WHC and texture properties of the adductor muscle. The morphology of ice crystals could be considerably modified by the AFP that was adsorbed to the ice crystal surface through hydrogen bonding with the US-VI, which significantly enhanced the WHC and texture properties of the scallop adductor muscle. Thus, some reference for investigating the US-VI and action mechanism of AFPs were provided in the frozen aquatic products. Subsequent research should focus on the effect of US-VI on protein structure changes, flavor, and nutritional value of the scallop adductor muscle. Moreover, given the current relatively high cost of commercial antifreeze agents, much work is still needed to achieve commercial manufacturing.

## Figures and Tables

**Figure 1 foods-11-00320-f001:**
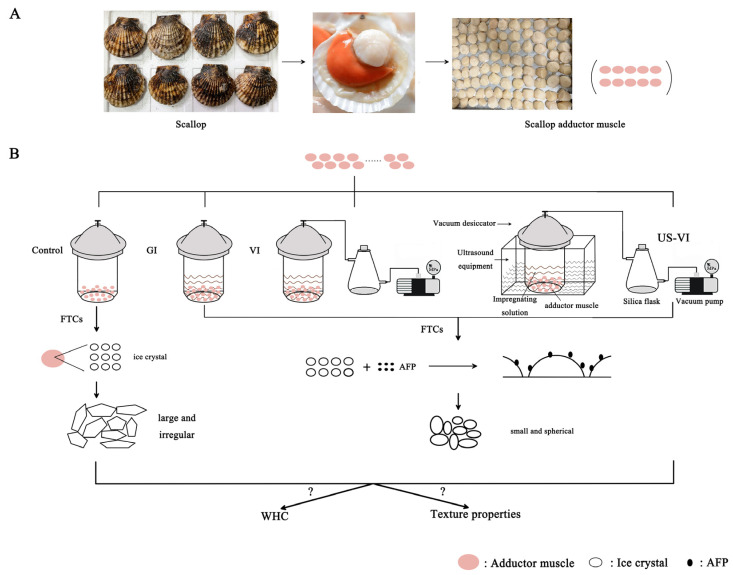
Schematic illustration for the experimental pre-treatment and potential mechanism. (**A**). The pre-treatment process of the experimental sample; (**B**). The Schematic illustration for the potential mechanism.

**Figure 2 foods-11-00320-f002:**
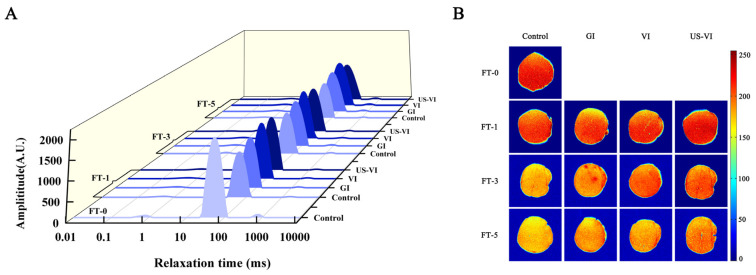
Changes in the water states and distribution of the scallop adductor muscle induced by freeze–thaw cycles. ((**A**) The T_2_ relaxation time curves of different water states; (**B**) T_2_ weighted magnetic resonance images; FT-0, 1, 3, and 5: number of freeze–thaw cycles.)

**Figure 3 foods-11-00320-f003:**
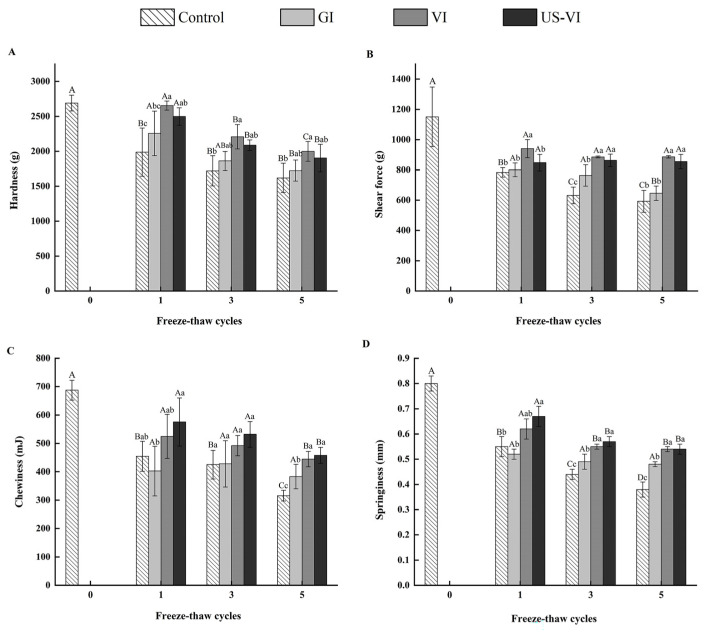
Changes in the texture of the scallop adductor muscle induced by freeze–thaw cycles. ((**A**) Hardness, (**B**) shear force, (**C**) chewiness, (**D**) springiness.) “A–D” indicated the difference in different number of FTCs (*p* < 0.05). “a–d” indicated the difference in samples with different pretreatment methods (*p* < 0.05).

**Figure 4 foods-11-00320-f004:**
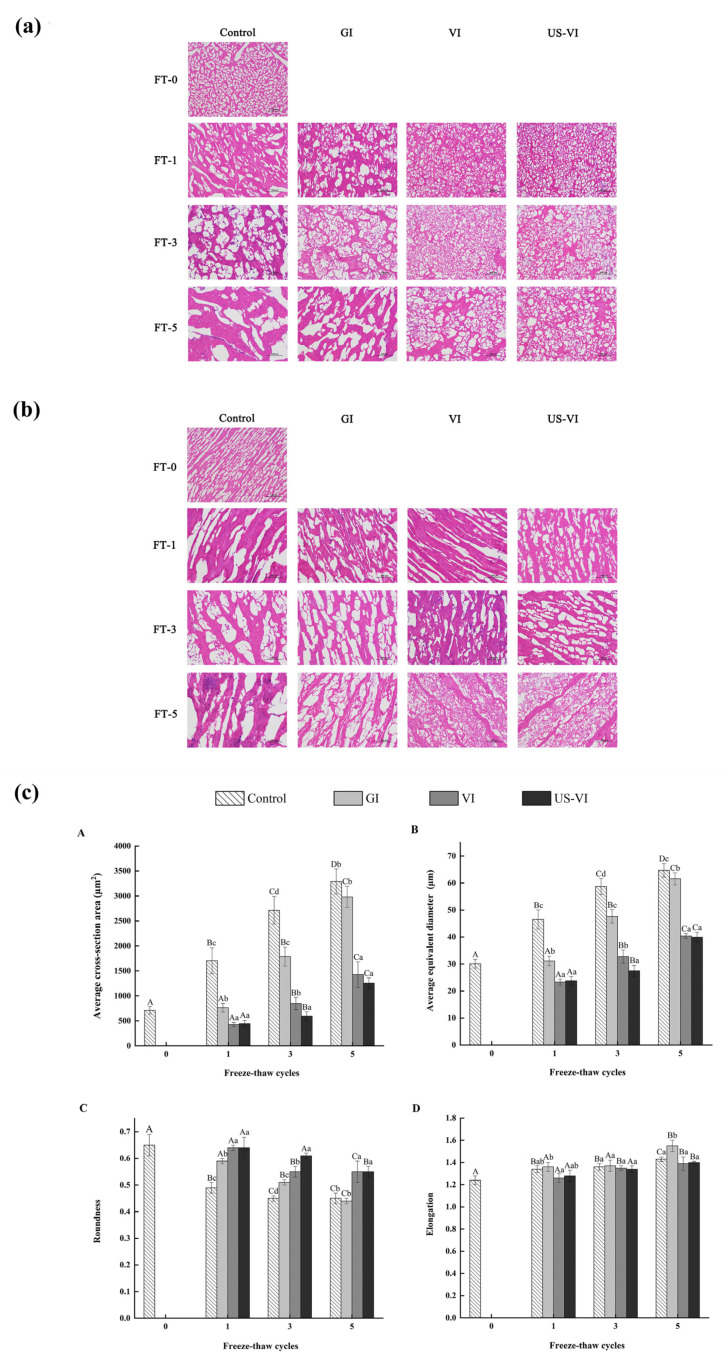
The observation and analysis of the ice crystal morphology of the scallop adductor muscle induced by freeze–thaw cycles (magnification: 200×). (**a**) Transverse section; (**b**) longitudinal section; (**c**) analysis of the ice crystals morphology ((**A**) average cross-section area; (**B**) average equivalent diameter; (**C**) roundness; (**D**) elongation). FT-0, 1, 3, and 5: number of freeze–thaw cycles. “A–D” indicated the difference in different number of FTCs (*p* < 0.05). “a–d” indicated the difference in samples with different pretreatment methods (*p* < 0.05).

**Figure 5 foods-11-00320-f005:**
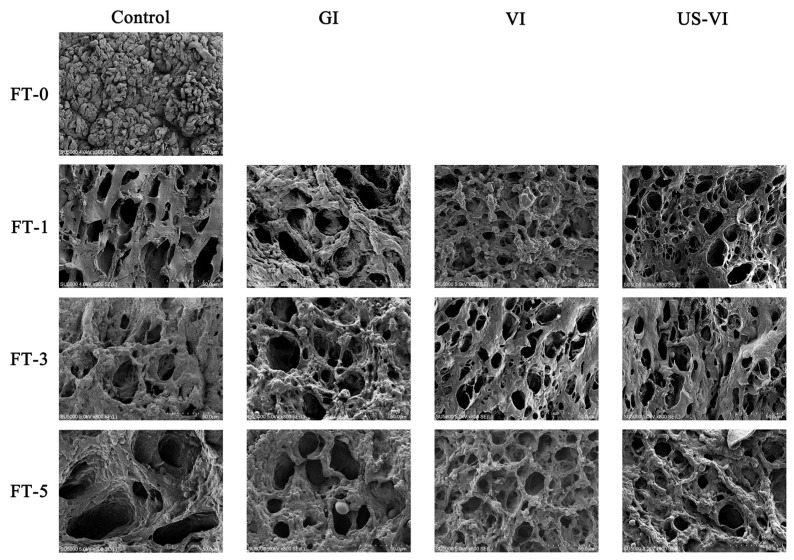
Changes in microstructural structure of the scallop adductor muscle induced by freeze–thaw cycles (magnification: 800×) (FT-0, 1, 3, and 5: numbers of freeze–thaw cycles).

**Figure 6 foods-11-00320-f006:**
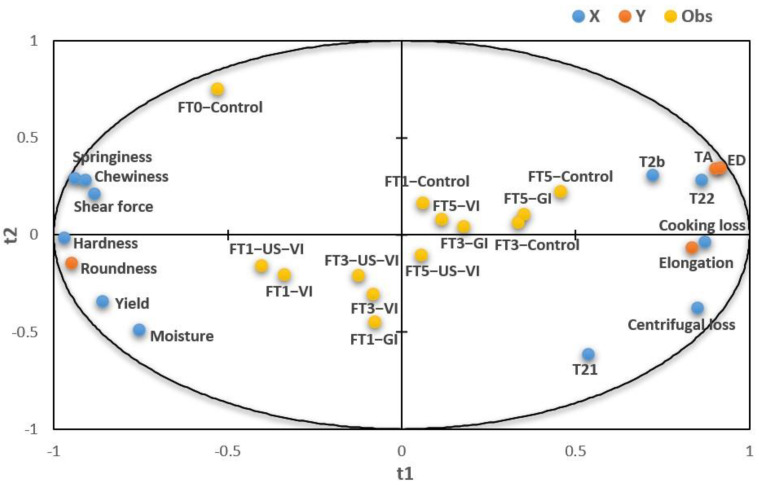
A correlation loading plot from a partial least squares regression (PLSR) model. (TA: average cross-sectional (transverse) area; ED: average equivalent diameter; FT1, FT3, FT5: numbers of freeze–thaw cycles.)

**Table 1 foods-11-00320-t001:** Changes in the water-holding capacity of the scallop adductor muscle induced by freeze–thaw cycles.

FTCs	Impregnation Method	Yield (%)	Moisture (%)	Centrifugal Loss (%)	Cooking Loss (%)	T_2_ Relaxation Times (ms)		
						T_2b_	T_21_	T_22_
**0**	**Fresh**	-	80.02 ± 0.28 ^A^	6.92 ± 0.51 ^A^	3.99 ± 0.94 ^A^	0.73 ± 0.07 ^A^	52.51 ± 0.54 ^A^	759.36 ± 4.51 ^A^
**1**	**Control**	98.81 ± 0.16 ^Ad^	78.52 ± 0.12 ^Bb^	27.98 ± 1.99 ^Bc^	16.41 ± 1.20 ^Bb^	0.71 ± 0.11 ^Aa^	69.05 ± 6.16 ^Ba^	1387.10 ± 100.83 ^Bc^
	**GI**	101.80 ± 0.74 ^Ac^	80.40 ± 0.15 ^Aa^	24.80 ± 2.48 ^Abc^	17.79 ± 2.57 ^Ab^	0.71 ± 0.06 ^Aa^	75.18 ± 7.46 ^Aa^	813.71± 96.27 ^Ab^
	**VI**	106.76± 0.58 ^Ab^	80.52 ± 0.38 ^Aa^	20.92 ± 0.69 ^Aab^	9.89 ± 2.90 ^Aa^	0.67 ± 0.09 ^Aa^	66.85 ± 5.29 ^Aa^	771.46 ± 141.98 ^Aab^
	**US-VI**	109.71± 0.58 ^Aa^	80.81 ± 0.16 ^Aa^	18.32 ± 0.96 ^Aa^	8.90 ± 1.16 ^Aa^	0.65 ± 0.04 ^Aa^	63.03 ± 4.58 ^Aa^	644.72 ± 38.81 ^Aa^
**3**	**Control**	94.06 ± 0.49 ^Bd^	78.74 ± 0.21 ^Bc^	35.27 ± 2.50 ^Cb^	34.41 ± 2.55 ^Cb^	0.84 ± 0.08 ^Ab^	73.42 ± 6.09 ^Bb^	1538.35 ± 0.00 ^Cc^
	**GI**	95.17 ± 0.91 ^Bc^	79.63 ± 0.58 ^Bb^	33.07 ± 2.80 ^Bb^	31.66 ± 1.42 ^Bab^	0.90 ± 0.00 ^Bb^	68.65 ± 2.52 ^Aab^	1019.21 ± 56.57 ^Bb^
	**VI**	98.10 ± 0.70 ^Bb^	80.52 ± 0.10 ^Aa^	33.73 ± 2.05 ^Bb^	28.14 ± 2.66 ^Ba^	0.73 ± 0.07 ^Aa^	65.33 ± 6.48 ^Aab^	838.60 ± 66.37 ^Ba^
	**US-VI**	101.01 ± 0.58 ^Ba^	80.18 ± 0.27 ^Aab^	29.04 ± 0.58 ^Ca^	25.68 ± 1.80 ^Ba^	0.70 ± 0.04 ^Aa^	65.19 ± 4.59 ^Aa^	803.18 ± 42.68 ^Ba^
**5**	**Control**	87.44 ± 0.35 ^Cd^	78.38 ± 0.23 ^Bc^	32.05 ± 2.66 ^Cb^	47.44 ± 1.87 ^Da^	0.82 ± 0.07 ^Aab^	65.81 ± 4.71 ^Ba^	1623.72 ± 145.30 ^Ca^
	**GI**	88.83 ± 0.55 ^Cc^	79.33 ± 0.13 ^Bb^	33.07 ± 0.88 ^Bb^	46.64 ± 2.75 ^Ca^	0.84 ± 0.06 ^Bb^	68.38 ± 3.74 ^Aa^	1615.71 ± 133.99 ^Ca^
	**VI**	92.77 ± 0.29 ^Cb^	79.22 ± 0.06 ^Bb^	32.35 ± 0.45 ^Bb^	42.06 ± 4.30 ^Ca^	0.71 ± 0.09 ^Aa^	62.01 ± 1.73 ^Aa^	1538.79 ± 73.90 ^Ca^
	**US-VI**	94.07 ± 0.07 ^Ca^	80.25 ± 0.23 ^Aa^	26.55 ± 0.72 ^Ba^	44.87 ± 3.25 ^Ca^	0.70 ± 0.05 ^Aa^	63.76 ± 2.67 ^Aa^	1437.52 ± 142.59 ^Ca^

Different letters (a–d, A–D) within the same row indicate significant differences (*p* < 0.05). “A–D” letters indicate significant differences in the same treatment groups during different freeze–thaw cycles, and letters “a–d” indicate significant differences in different treatment groups under the same freeze–thaw cycles. Note: FTCs: freeze–thaw cycles; T_2b_: bound water; T_21_: immobilized water; T_22_: free water; GI: general impregnation; VI: vacuum impregnation; US-VI: ultrasound-assisted vacuum impregnation.

## Data Availability

The data presented in this study are available in the article.

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
