# Peer review of "Effects of Ultrasound-Assisted Vacuum Impregnation Antifreeze Protein on the Water-Holding Capacity and Texture Properties of the Yesso Scallop Adductor Muscle during Freeze–Thaw Cycles"

_foods, 2022, doi:10.3390/foods11030320_

Round 1
Reviewer 1 Report
Shi et al
The manuscript is generally well written and clear. It will make an important contribution to the literature when published. There are some minor english corrections needed in the introduction and the abstract but the rest of the manuscript seems okay
Corrections.
line 30 delete “To explore”
line 31 change to read “…(FTCs) were evaluated based on…”
line 32: start a new sentence after (US-VI)
line 36 define T22 also best to more clearly define T2 and T21 etc… in the manuscript
line 38: delete second “the”
line 50: change to read “…show great profitability but are highly…”
line 52: delete “and then maintain” add “if”
line 52/53: delete “to the greatest extent” add “are maintained”
line 65: change second “crystals” to “crystal”
line 73: delete “more”
line 74: delete “rare” and “have been”
line 75 add “are rare” at end of sentence
line 83: scarce
Author Response
Dear reviewer and editor:
First and foremost, I’d like to express my gratitude for your approval of my manuscript, which is really valuable to me. Second, thank you for improving the professionalism of my article by correcting my English expression. I have made careful modifications on the original manuscript in response to these comments and suggestions. All changes were highlighted in red. Our point-by-point responses to your comments are listed below:
- line 30-delete “To explore”: I have deleted this in Line 30.
- line 31-change to read “…(FTCs) were evaluated based on…”: I have changed this in Line 31.
- line 32-start a new sentence after (US-VI): I have made change in Line 33.
- line 36-define T22 also best to more clearly define T2 and T21 etc… in the manuscript: T2, T2b, T21 and T22 were explained in detail in line 150 and line 228-231 of text. T2 represents the transverse relaxation time. T2b (0-10 ms), the proton pool with the shortest relaxation time, represents bound water attached on the polar groups of macromolecules directly. T21 (10-100 ms) and T22 (100-1000 ms) were assigned to immobilized water entrapped within the myofibrillar network and ascribed to free water in the myofibril lattice, respectively.
- line 38: delete second “the”: I have changed this in Line 38.
- line 50: change to read “…show great profitability but are highly…”: I have modified the sentence according to this pattern in Line 52.
- line 52: delete “and then maintain” add “if” and delete “to the greatest extent” add “are maintained: I am sorry to say that I don’t think “if” can convey the meaning of sentence very well. This sentence is meant to convey that “freezing could prolong the shelf-life of products and maximize the preservation of their original properties by inhibiting microbial activity and enzymatic reactions”. So I did another change in Line 53-54.
- line 65: change second “crystals” to “crystal”: I have changed this in Line 69.
- line 73: delete “more”: I have changed this in Line 78.
- line 74: delete “rare” and “have been” and add “are rare” at end of sentence in Line 75: I have modified the sentence according to this pattern in Line 79.
- line 83: scarce: I am sorry for these spelling mistake, I’ve rechecked the article.
These are the results of my modifications. Please contact me again if you have any questions, and we will respond to your remarks as soon as possible. We hope that these revisions are satisfactory and that the revised version will be acceptable for publication in Foods.
Thank you very much for your work concerning my paper.
Best wishes,
Yuyao Shi and Xichang Wang.
Reviewer 2 Report
The quality of the work is very good but wondering about the practicality for commercial production. For example where do the infused cryo-protective proteins come from? Are they commercially available? If so, does the cost of the technology justify the yield and quality increases?
Author Response
Dear reviewer and editor:
First and foremost, I’d like to express my gratitude for your approval of my manuscript, which is really valuable to me.
Thank you for your questions regarding the practicality on commercial production, which is one of the most pressing concerns in the industrial application of antifreeze proteins. The AFP used in this investigation was commercially available, which was noted in 2.1 Sample preparation (Line 98). Although the current relatively high cost of commercial antifreeze agents, it has been confirmed that AFP can be extracted from herring [1], wheat, tilapia scales [2], pigskin collagen [3], fish collagen [4] and other plants and animals of the alpine zone, which could inhibit quality deterioration of products. Therefore, the reduction of AFP cost and commercial production is a future development trend, which will benefit the frozen food industry. Moreover, another highlight of this study is the impregnation method of US-VI. According to the previous studies, it is known that AFP is most commonly added by mixing it with liquid or injecting it into muscle tissue. Injections in aquatic products can directly impair tissue integrity and are added in inconsistent volumes. As a result, we attempted to use the method of US-VI to add antifreeze agents with a larger molecular weight into the tissue in order to make better use of the AFP.
These are the results of my modifications. Please contact me again if you have any questions, and we will respond to your remarks as soon as possible. We hope that these revisions are satisfactory and that the revised version will be acceptable for publication in Foods.
Thank you very much for your work concerning my paper.
Best wishes,
Yuyao Shi and Xichang Wang.
- Nian, L.; Cao, A.; Cai, L. Investigation of the antifreeze mechanism and effect on quality characteristics of largemouth bass (Micropterus salmoides) during F-T cycles by hAFP. Food Chemistry 2020, 325, 126918, DOI:https://doi.org/10.1016/j.foodchem.2020.126918.
- Chen, X.; Li, L.; Yang, F.; Wu, J.; Wang, S. Effects of gelatin-based antifreeze peptides on cell viability and oxidant stress of Streptococcus thermophilus during cold stage. Food and Chemical Toxicology 2020, 136, 111056, DOI:https://doi.org/10.1016/j.fct.2019.111056.
- Cao, H.; Zhao, Y.; Zhu, Y.B.; Xu, F.; Yu, J.S.; Yuan, M. Antifreeze and cryoprotective activities of ice-binding collagen peptides from pig skin. Food Chemistry 2016, 194, 1245-1253, DOI:https://doi.org/10.1016/j.foodchem.2015.08.102.
- Damodaran, S.; Wang, S. Ice crystal growth inhibition by peptides from fish gelatin hydrolysate. Food Hydrocolloids 2017, 70, 46-56, DOI:https://doi.org/10.1016/j.foodhyd.2017.03.029.
Reviewer 3 Report
The manuscript entitled “Effects of ultrasound-assisted vacuum impregnation antifreeze protein on the water-holding capacity and texture properties of the Yesso scallop adductor muscle during freeze-thaw cycles” presents information in regards to Yesso scallop processing
- Authors should follow the authors guidelines.
- In this section Authors presented the information associated with the processing of scallops, its quality and mechanical damage to the muscle fibers (due to the freezing). This section should be presented – what do we know and what is the background for this study. Some detailed information about other studies are necessary (international context – the situation in other countries should be presented). The good background should present the history of problem, the current knowledge and scientific "gap", and then authors should present how their study could fill this gap to justify the study.
- ‘2.2.3 Texture profile analysis (TPA) and shear force’ – pleas add the information about parameters (with units)
- Number of replication should be provided for each measurements (specify it)
- Line 181 – ‘mean values ± standard deviations’ – Was the normality of distribution tested? The information about it should be added and authors should be consequent. If data have normal distribution, they should be treated as such, if not, nonparametric tests should be applied. Please specify it.
- The table must stand alone (should be understandable without referring to the text – including abbreviations which should be explained as footnotes)
- Data for texture (Changes in the texture) should be presented in tables (in figure it is difficult to follow)
- Figure 4 is tool small and poor resolution
- Figures 6 must be better describe (in the presented form is not understandable enough)
- Authors should in their discussion include 3 areas: (1) compare gathered data with the results by other authors, (2) formulate implications of the results of their study and studies by other authors, (3) formulate the future areas which should be studied.
- Authors should present here and discuss the limitations of their study.
- There are typos in manuscript (e.g. ‘(43 kHz ,200 w)’) - please correct it
Author Response
Dear reviewer and editor:
We’d like to express my gratitude for your professional comments and thoughtful suggestions, which is very useful for me. We have made careful modifications on the original manuscript in response to these comments and suggestions. in response to these comments and suggestions. All changes were highlighted in red. In addition, we have consulted native English speakers for paper revision before the submission this time. Our point-by-point responses to your comments are listed below:
Comment 1: In this section Authors presented the information associated with the processing of scallops, its quality and mechanical damage to the muscle fibers (due to the freezing). This section should be presented – what do we know and what is the background for this study. Some detailed information about other studies are necessary (international context – the situation in other countries should be presented). The good background should present the history of problem, the current knowledge and scientific "gap", and then authors should present how their study could fill this gap to justify the study.
Answer: Thank you very much for your comments. The international context of scallops was added in lines 47-49. I think the framework you presented was already addressed in the section of Introduction, and I added some new content. First, the history of problem: 1. The production of scallops showed an increasing trend, but they have a short shelf-life. Scallops are mostly distributed as the form of frozen products (Line 52-54). 2. Long-term frozen storage inevitably cause quality deterioration of aquatic products, so they need to be quality-controlled (Line 56-61). Second, the current knowledge: 1. Compared with other common antifreeze agents, AFP had better frozen stability and did not affect its taste and flavor (New Additions in Line 65); 2. Vacuum impregnation and Ultrasound treatment can increase the permeability of muscle tissues through cavitation forces and accelerated the rate of mass transfer. They have been applied to introduce some additives into the tissue (Line 76-86). Finally, Scientific "gap": 1. Appropriate external forces are required to further dispersion and permeation into the tissue due to the large molecular weight of AFP, and little attention has been paid to the AFP on the WHC and texture properties of muscle (Line 71-75). 2. There has been scarce information on the effects of ultrasound combined with vacuum impregnation AFP, especially the comprehensive investigation of quality on the scallop adductor muscle after the combination (Line 87-89). Thus, based on the US-VI, the effect of AFP on the WHC and texture properties of the scallop adductor muscle were investigated. Some reference for the quality control of frozen aquatic products under the effect of the AFP during storage could be provided. If there is still something that not be considered, please point it out to me and I will fix them carefully.
Comment 2: ‘2.2.3 Texture profile analysis (TPA) and shear force’ – pleas add the information about parameters (with units)
Answer: I have added the units of parameters accordingly and marked them in section 2.2.3 (Line 139-143).
Comment 3: Number of replication should be provided for each measurements (specify it)
Answer: In 2.2.7 Statistical analysis, I have indicated that “All experiments in this study were performed at least in triplicate.” Experiments with more than three repetitions have been specifically indicated in the text. (Line 120, 144 and 165)
Comment 4: Line 181 – ‘mean values ± standard deviations’ – Was the normality of distribution tested? The information about it should be added and authors should be consequent. If data have normal distribution, they should be treated as such, if not, nonparametric tests should be applied. Please specify it.
Answer: Yes, we tested that all data conformed to the normal distribution using the SPSS 21.0. Only data that are continuous variables and conform to the normal distribution can be expressed by the form of ‘mean values ± standard deviations’ and analyzed for significance. Additional clarification was provided in line 188 of the text. (Further explanation is provided in the attachment.)
Comment 5: The table must stand alone (should be understandable without referring to the text – including abbreviations which should be explained as footnotes.
Answer: In the table footnotes, we've included the definitions of all abbreviations that appear in the table 1 (Line 220-221, 257, 347 and 379).
Comment 6: Figure 4 is tool small and poor resolution
Answer:Figure 4 has been reinserted into Line 318, enlarged and higher resolution.
Comment 7: Figures 6 must be better describe (in the presented form is not understandable enough)
Answer:Figure 6 has been explained in detail in Line 350-363.
Comment 8: Authors should in their discussion include 3 areas: (1) compare gathered data with the results by other authors, (2) formulate implications of the results of their study and studies by other authors, (3) formulate the future areas which should be studied.
Answer:We already have some sections for comparing the collected data with the results of other authors in the original manuscript (Line 206-207, 234-235, 270-271 and 291-292). We also added some new comparisons (Line 312-314 and 336-337). The line 314-316 of text highlights the importance of this study in comparison to other relevant investigations. The line 388-392 of text formulated the future areas which should be studied.
Comment 9: Authors should present here and discuss the limitations of their study.
Answer:Given the current relatively high cost of commercial antifreeze agents, much work is still needed to achieve commercial manufacturing (Line 391-392). Although the current relatively high cost of commercial antifreeze agents, it has been confirmed that AFP can be extracted from herring [1], wheat, tilapia scales [2], pigskin collagen [3], fish collagen [4] and other plants and animals of the alpine zone, which could inhibit quality deterioration of products. Therefore, the reduction of AFP cost and commercial production is a future development trend, which will benefit the frozen food industry.
Comment 10: There are typos in manuscript (e.g. ‘(43 kHz ,200 w)’) - please correct it
Answer:Thank you for your careful revision comments. I have changed ‘43 kHz ,200 w’ to ‘43 kHz, 200 W/cm2 (Line 104).
